# Maternal Death Related to Sudden Unexpected Death in Epilepsy: A Nationwide Survey in Japan

**DOI:** 10.3390/brainsci11080995

**Published:** 2021-07-28

**Authors:** Hiroaki Tanaka, Shinji Katsuragi, Junichi Hasegawa, Kayo Tanaka, Masamitsu Nakamura, Eijiro Hayata, Masahiko Nakata, Akihiko Sekizawa, Isamu Ishiwata, Tomoaki Ikeda

**Affiliations:** 1Department of Obstetrics and Gynecology, Mie University School of Medicine, 2-174 Edobashi, Tsu 514-8507, Japan; skatsura@clin.medic.mie-u.ac.jp (S.K.); tagami.t.ky@gmail.com (K.T.); t-ikeda@clin.medic.mie-u.ac.jp (T.I.); 2Department of Obstetrics and Gynecology, St. Marianna University School of Medicine, 2-16-1 Sugao, Miyamae, Kawasaki 216-8511, Japan; hasejun@marianna-u.ac.jp; 3Department of Obstetrics and Gynecology, Showa University School of Medicine, 1-5-8 Hatanodai, Shinagawa, Tokyo 142-8666, Japan; nakamula-m@med.showa-u.ac.jp (M.N.); sekizawa@med.showa-u.ac.jp (A.S.); 4Department of Obstetrics and Gynecology, Toho University Omori Medical Center, 6-11-1 Omorinishi, Ota, Tokyo 143-8540, Japan; e_hayata@hotmail.com (E.H.); masahiko.nakata@med.toho-u.ac.jp (M.N.); 5Department of Obstetrics and Gynecology, Ishiwata Hospital, 1-4-21 Kamimito, Mito 310-0041, Japan; mismtiis@mito.ne.jp

**Keywords:** maternal death, pregnancy, sudden unexpected death in epilepsy, epilepsy

## Abstract

Sudden unexpected death in epilepsy (SUDEP) is defined as the sudden death of a patient with epilepsy in the absence of an anatomic or toxicologic cause. Whether pregnancy is a risk factor for SUDEP is unclear. Using data submitted to the Japan Association of Obstetricians and Gynecologists (JAOG), which has been collating information regarding all maternal deaths in Japan since 2000, this study evaluated maternal mortality data from 2010 to 2019 to evaluate the current circumstances of maternal death related to SUDEP in Japan. Six women died due to SUDEP during this period; the maternal mortality rate related to SUDEP was 0.066/100,000 individuals. Two women each died during the second trimester, third trimester, and postpartum period. Four and two women were receiving monotherapy and no therapy with anti-epileptic drugs, respectively. The duration of epilepsy was ≤15 years in three women, >15 years in one woman, and unknown in two women. This study furthers our understanding of the prevalence of maternal deaths due to SUDEP in Japan. Further studies are needed to confirm whether pregnancy is a risk factor for SUDEP.

## 1. Introduction

Sudden unexpected death in epilepsy (SUDEP) is defined as the sudden death of a patient with epilepsy in the absence of an anatomic or toxicologic cause [1]. In recent years, SUDEP has received increasing attention because the incidence of sudden death in patients with epilepsy is approximately 20 times higher than that of the general population [2]. Several mechanisms for SUDEP have been suggested including hypoventilation due to an epileptic attack, arrhythmia, central neural dysfunction, and autonomic nervous system dysfunction [3,4,5,6,7,8,9]. Furthermore, high frequency of tonic-clonic convulsions, male sex, long duration of epilepsy, and multidrug therapy have been proposed as risk factors for SUDEP [10,11,12,13,14,15,16,17,18,19,20,21]. However, it is unclear whether pregnancy is a risk factor for SUDEP.

The prevalence of epilepsy is 8 in 1000, and it is estimated to affect over 1,000,000 patients in Japan [22]. Therefore, a substantial number of patients with epilepsy become pregnant. The aim of this study was to investigate the current circumstances of maternal death related to SUDEP in Japan.

## 2. Materials and Methods

Information regarding all maternal deaths in Japan has been gathered by the Japan Association of Obstetricians and Gynecologists (JAOG) since 2010. The details of all maternal deaths in Japan are submitted to the JAOG, and the individual data is analyzed by the Maternal Death Exploratory Committee (Chairman: Ikeda, T.). This committee consists of 15 obstetricians, four anesthesiologists, two pathologists, an emergency physician, and various specialists who attend the review sessions each month and make annual recommendations aimed at reducing the maternal mortality rate. The present study was performed by this committee as part of a series of analyses on maternal deaths in Japan, using maternal mortality data from 2010 to 2019. We investigated the rate of maternal deaths related to epilepsy in Japan, and maternal death was defined as maternal death from pregnancy to 42 days postpartum.

The diagnosis of definite SUDEP was made when the following conditions were met: (1) a history of epilepsy (more than one epileptic attack within five years), (2) sudden death, (3) unexpected death (death in the absence of an apparent associated disease), and (4) unexplained death despite a thorough investigation of the causes of sudden death [1]. A diagnosis of probable SUDEP was made when criteria of (1)–(3) but not (4) were met [1].

The following demographic and diagnosis-related data were collected for each patient, namely maternal background; age; parity; diagnosis of SUDEP; periods of death (trimester of pregnancy or postpartum); antiepileptic drug (AED) therapy including lamotrigine therapy; epilepsy attacks; annual frequency of generalized tonic-clonic seizure (GTCS) over the preceding three years and during pregnancy; presence of a bystander at the time of death, duration of epilepsy; and autopsy results.

This study was approved by the ethics committee of the National Cerebral and Cardiovascular Center of Japan under the title “Research on a model project regarding surveys and evaluations on maternal mortality in Japan” (receipt number N18-34).

## 3. Results

The total number of maternal deaths between January 2010 and December 2018 in Japan was 407. Maternal death related to epilepsy occurred in seven women: six due to SUDEP and one due to seizures. The combined number of all pregnancies resulting in live birth or fetal deaths in Japan during the study period was 9,039,897 pregnancies. The total number of maternal deaths related to SUDEP was 6/9,039,897 pregnancies. The maternal mortality rate related to SUDEP was 0.066/100,000 pregnancies. The mortality related to epilepsy for trienniums between 2010 and 2018 is shown in Table 1.

The characteristics of maternal death in the six cases related to SUDEP are shown in Table 2. Four and two women met the criteria for probable and definite SUDEP, respectively. The periods of death were the second trimester in two women, third trimester in two women, and postpartum in two women. Four and two women were receiving AED monotherapy and no AED therapy, respectively. One patient was taking lamotrigine. The duration of epilepsy was ≤15 years in three women, >15 years in one woman, and unknown in two women. GTCS frequency per year (more than three times per year) was around 1 year of pregnancy.

## 4. Discussion

This study analyzed maternal mortality associated with epilepsy in Japan. We clarified the total number of deaths from epilepsy and SUDEP during pregnancy in Japan, and demonstrated that most epilepsy-related maternal deaths were SUDEP rather than convulsions.

According to the Center for Maternal and Child Enquiries in the United Kingdom (UK), the maternal mortality rate for epilepsy from 1991 to 2008, calculated on a 3-year basis, was 0.39–0.86; SUDEP accounted for about half of all epilepsy-related maternal deaths. The rates of epilepsy-related maternal mortality in Japan are much lower than those in the UK, and unlike the UK, most epilepsy-related maternal deaths in Japan were SUDEP. These differences in epilepsy-related maternal mortality between Japan and other countries need to be clarified in future studies; one possible explanation may be the lower pregnancy rates among women with severe or uncontrolled epilepsy in Japan.

Understanding the mechanism of SUDEP is the key to prevention. Previous studies have demonstrated that AED multitherapy, chronic epilepsy, lamotrigine therapy, an epileptic heart (heart and coronary vasculature damaged by chronic epilepsy as a result of re-peated surges in catecholamines and hypoxemia, leading to electrical and mechanical dysfunction), and frequent GTCS are risk factors for SUDEP [23,24]. A recently published article has shown that cardiac iron accumulation can trigger ferroptosis and be the cause of SUDEP [25]. However, we did not find a clear association between these factors and SUDEP in this study. Therefore, it is plausible that there are currently unidentified risk factors for SUDEP, which need to be identified through future studies in order to formulate effective preventive strategies.

The general incidence of SUDEP in epilepsy patients has been reported to be 0.1–1/1000 per year [25]. In Japan, there is no national registry of pregnant women with epilepsy, and the rates of SUDEP among all pregnant women is unknown and it is important to clarify the total number of SUDEP cases during pregnancy in Japan. Therefore, it remains to be verified whether pregnancy is a risk factor for SUDEP, especially considering the fact that many women with well-controlled epilepsy can have healthy pregnancies and previous studies have established that SUDEP is more likely to occur when chronic control is poor [26]. However, as shown in this study, patients with well-controlled epilepsy, including those with relatively short durations of epilepsy, can also develop SUDEP during pregnancy, and the mechanism and triggers have not been clarified. This study has some limitations worth noting including its retrospective design. Future prospective studies are warranted to confirm the association of SUDEP with pregnancy in Japan.

## 5. Conclusions

We clarified the prevalence of maternal deaths related to epilepsy in Japan and found that most maternal deaths associated with epilepsy were SUDEP. Notably, we found that women without risk factors for SUDEP developed SUDEP during pregnancy. This highlights the need for further studies to clarify the pathomechanism of SUDEP to develop preventative strategies.

## Figures and Tables

**Table 1 brainsci-11-00995-t001:** The mortality related to epilepsy for the trienniums between and including 2010–2018 in Japan.

	Numbers of Pregnancies	Number of Maternal Deaths	Number of Deaths Related to Epilepsy	Maternal Death Rate Related to Epilepsy
2010–2012	3,159,341	146	0	0
2013–2015	3,039,032	133	5	0.16
2016–2018	2,841,524	128	2	0.07

**Table 2 brainsci-11-00995-t002:** The characteristics of maternal death cases related to SUDEP.

No.	Maternal Age	Primipara	Diagnosis	Period of Death	AED Therapy	Autopsy	Lamotrigine Therapy	Duration of Epilepsy	GTCS Frequency Per Year
1	40s	Yes	Probable SUDEP	Postpartum	No AED therapy	No	No	Unknown	No
2	20s	Yes	Definite SUDEP	Postpartum	Monotherapy	Yes	No	≤15 years	No
3	30s	No	Probable SUDEP	Second trimester	Monotherapy	No	No	Unknown	No
4	40s	No	Probable SUDEP	Second trimester	No AED therapy	No	No	≤15 years	3<
5	30s	Yes	Definite SUDEP	Third trimester	Monotherapy	Yes	Yes	>15 years	No
6	20s	Yes	Probable SUDEP	Third trimester	Monotherapy	No	No	≤15 years	No

SUDEP, sudden unexpected death in epilepsy; AED, anti-epileptic drug; GTCS, generalized tonic-clonic seizure.

## Data Availability

The data presented in this study are available on request from the corresponding author.

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
