# Peer review of "Maternal Death Related to Sudden Unexpected Death in Epilepsy: A Nationwide Survey in Japan"

_brainsci, 2021, doi:10.3390/brainsci11080995_

Round 1

Reviewer 1 Report

In this work, the authors present a summary of deaths in epilepsy, mainly SUDEP, associated with maternity. The work is correctly presented but some minor improvements need to be made.
In this sense, they should redraw Table 2 to avoid incorrect separation of the words.
A recently published basic article has shown that cardiac iron accumulation can trigger ferroptosis and be the cause of SUDEP. The authors should include this article as a possible cause of SUDEP in the introduction as they report postpartum deaths with no frequency from GTCS.
doi: 10.3389 / fneur.2021.609236
Based on the reported results, the authors should discuss other possible causes of SUDEP (such as cardiac dysfunction) that make it possible to postulate pregnancy as a risk factor even in controlled epilepsy.

Author Response

Reviewer 1

In this work, the authors present a summary of deaths in epilepsy, mainly SUDEP, associated with maternity. The work is correctly presented but some minor improvements need to be made.
In this sense, they should redraw Table 2 to avoid incorrect separation of the words.
A recently published basic article has shown that cardiac iron accumulation can trigger ferroptosis and be the cause of SUDEP. The authors should include this article as a possible cause of SUDEP in the introduction as they report postpartum deaths with no frequency from GTCS.
doi: 10.3389 / fneur.2021.609236
Based on the reported results, the authors should discuss other possible causes of SUDEP (such as cardiac dysfunction) that make it possible to postulate pregnancy as a risk factor even in controlled epilepsy.

  • We have revised Table 2 as suggested by reviewer 1.
  • We have added the epileptic heart and cardiac iron accumulation in discussion section.

Reviewer 2 Report

This well-written paper by Tanaka et al. examines maternal death rates in Japan due to SUDEP. Whether pregnancy affects SUDEP risk is an important clinical question that has not been explored in much detail. Surveying all pregnancies between 2010-18, this article measured the maternal death rate due to SUDEP across Japan and then gives important demographic data for the 6 SUDEP cases that were identified. Although brief, this paper provides some key demographic survey data for SUDEP related to pregnancy. However, there are some relatively minor concerns that could be better addressed to further strengthen the paper.

CONCERNS:

On page 2, line 74 it says “the combined number of all live births and fetal deaths…”. Shouldn’t this instead say “the combined number of all pregnancies resulting in live birth or fetal deaths…”?

The mortality rate was calculated as the number of SUDEP deaths per 100,000 individuals. However, this seems like it should more precisely be the rate per 100,000 pregnancies since some individuals likely had more than 1 pregnancy during that period. This change from “individuals” to “pregnancies” would seem consistent with the labeling in Table 1.

Since there is no national registry of pregnant women with epilepsy in Japan, the maternal SUDEP rate among people with epilepsy cannot apparently be calculated. However, is it possible to do a best approximation based on the epilepsy prevalence of 8/1000 given in the intro section? If this were the approximate rate this would mean about 1200 mothers in the data set had epilepsy which would give a SUDEP rate of about 0.8/1000 which is similar to other populations. If this approximated rate is not obviously higher than other population estimates, this would seem to imply there is likely not a drastic difference in SUDEP risk due to pregnancy.

The spacing of the words in Table 2 needs to be corrected so that they fit within the columns correctly.

Author Response

Reviewer 2

This well-written paper by Tanaka et al. examines maternal death rates in Japan due to SUDEP. Whether pregnancy affects SUDEP risk is an important clinical question that has not been explored in much detail. Surveying all pregnancies between 2010-18, this article measured the maternal death rate due to SUDEP across Japan and then gives important demographic data for the 6 SUDEP cases that were identified. Although brief, this paper provides some key demographic survey data for SUDEP related to pregnancy. However, there are some relatively minor concerns that could be better addressed to further strengthen the paper.

CONCERNS:

On page 2, line 74 it says “the combined number of all live births and fetal deaths…”. Shouldn’t this instead say “the combined number of all pregnancies resulting in live birth or fetal deaths…”?

  • We have revised the sentence as follow; “the combined number of all pregnancies resulting in live birth or fetal deaths…”

The mortality rate was calculated as the number of SUDEP deaths per 100,000 individuals. However, this seems like it should more precisely be the rate per 100,000 pregnancies since some individuals likely had more than 1 pregnancy during that period. This change from “individuals” to “pregnancies” would seem consistent with the labeling in Table 1.

  • We have revised the sentence from individuals to pregnancies.

Since there is no national registry of pregnant women with epilepsy in Japan, the maternal SUDEP rate among people with epilepsy cannot apparently be calculated. However, is it possible to do a best approximation based on the epilepsy prevalence of 8/1000 given in the intro section? If this were the approximate rate this would mean about 1200 mothers in the data set had epilepsy which would give a SUDEP rate of about 0.8/1000 which is similar to other populations. If this approximated rate is not obviously higher than other population estimates, this would seem to imply there is likely not a drastic difference in SUDEP risk due to pregnancy.

  • We think that the maternal SUDEP rate among people with epilepsy have to show in this manuscript. However, there is no national registry of pregnant women with epilepsy in Japan as suggested by reviewer. We think that approximation based on the epilepsy prevalence of 8/1000 may misguide the true the maternal SUDEP rate among people with epilepsy Therefore, we have not wrikte the approximation based on the epilepsy prevalence of 8/1000.

The spacing of the words in Table 2 needs to be corrected so that they fit within the columns correctly.

  • We have revised Table 2.
